# Impact of Exogenously Sprayed Antioxidants on Physio-Biochemical, Agronomic, and Quality Parameters of Potato in Salt-Affected Soil

**DOI:** 10.3390/plants11020210

**Published:** 2022-01-14

**Authors:** Eman Selem, Asem A. S. A. Hassan, Mohamed F. Awad, Elsayed Mansour, El-Sayed M. Desoky

**Affiliations:** 1Botany and Microbiology Department, Faculty of Science, Zagazig University, Zagazig 44519, Egypt; eman.elsaid@zu.edu.eg; 2Horticulture Department, Faculty of Agriculture, Zagazig University, Zagazig 44519, Egypt; asem.ahmed92@gmail.com; 3Department of Biology, College of Science, Taif University, P.O. Box 11099, Taif 21944, Saudi Arabia; m.fadl@tu.edu.sa; 4Department of Crop Science, Faculty of Agriculture, Zagazig University, Zagazig 44519, Egypt; 5Botany Department, Faculty of Agriculture, Zagazig University, Zagazig 44519, Egypt; Sayed1981@zu.edu.eg

**Keywords:** salinity, folic, ascorbic, salicylic, agronomic traits, physiological attributed, biochemical parameters, tuber yield, tuber quality

## Abstract

Salinity is one of the harsh environmental stresses that destructively impact potato growth and production, particularly in arid regions. Exogenously applied safe–efficient materials is a vital approach for ameliorating plant growth, productivity, and quality under salinity stress. This study aimed at investigating the impact of foliar spray using folic acid (FA), ascorbic acid (AA), and salicylic acid (SA) at different concentrations (100, 150, or 200 mg/L) on plant growth, physiochemical ingredients, antioxidant defense system, tuber yield, and quality of potato (*Solanum tuberosum* L cv. Spunta) grown in salt-affected soil (EC = 7.14 dS/m) during two growing seasons. The exogenously applied antioxidant materials (FA, AA, and SA) significantly enhanced growth attributes (plant height, shoot fresh and dry weight, and leaves area), photosynthetic pigments (chlorophyll *a* and *b* and carotenoids), gas exchange (net photosynthetic rate, *Pn*; transpiration rate, *Tr*; and stomatal conductance, *gs*), nutrient content (N, P, and K), K^+^/ Na^+^ ratio, nonenzymatic antioxidant compounds (proline and soluble sugar content), enzymatic antioxidants (catalase (CAT), peroxidase (POX), superoxide dismutase (SOD), and ascorbate peroxidase (APX)) tuber yield traits, and tuber quality (dry matter, protein, starch percentage, total carbohydrates, and sugars percentage) compared with untreated plants in both seasons. Otherwise, exogenous application significantly decreased Na^+^ and Cl^−^ compared to the untreated control under salt stress conditions. Among the assessed treatments, the applied foliar of AA at a rate of 200 mg/L was more effective in promoting salt tolerance, which can be employed in reducing the losses caused by salinity stress in potato grown in salt-affected soils.

## 1. Introduction

Potato (*Solanum tuberosum* L.) is an essential food crop either for local consumption or export. It has essential beneficial values in human nutrition as an adequate source of starch, energy, vitamins, minerals, and organic acids [1]. Potato is considered the fourth essential crop in the world after wheat, rice, and maize [2]. Its total cultivated area is about 17.34 million hectares, which produce almost 370.43 million tons annually [2].

Salinity is one of the harsh environmental stresses that destructively impact potato growth and production, particularly in arid regions [3]. Large areas of cultivated soils are salt affected, mainly in arid regions due to high evaporation, low precipitation, poor drainage, poor irrigation practices, rising water tables, or using saline water in irrigation [4,5].

Plants grown in salt-affected soils suffer from decreased water availability, which causes water shortfall, specific toxic impacts of Na^+^ and Cl^−^ ions, and nutrient imbalance by reducing the absorption of certain elements [6,7]. Salt stress obstructs the water relations of the plant, which results in osmotic stress [8]. Additionally, accumulation of salt in the leaf apoplast causes turgor loss and dehydration, which leads to the death of tissues and cells [9]. Under salt stress, there is a decrease in or inhibition of nutrient uptake, which leads to a lack of chlorophyll content [10,11]. Moreover, salinity causes a considerable reduction in plant growth by inhibiting plant metabolism [12,13]. It destructively impacts plant metabolism through osmotic stress and high production of ROS such as O_2_^−^, OH^−^, and H_2_O_2_ [14].

Potato, like other field crops, is influenced by environmental stresses such as soil salinity, drought stress, and other biological and physical effects, especially under current climate change [15]. Foliar feeding can be applied successfully to compensate for the shortage of certain elements and enhance plant growth, particularly under environmental stresses [12,16]. The present global scenario strongly affirms the importance of sustainable agriculture and eco-friendly agricultural practices [17]. Recently, certain safe–efficient products have been employed to mitigate the environmental stresses on field crops such as folic acid (vitamin B9), ascorbic acid (vitamin C), and salicylic acid.

Folic acid (FA) is a natural antioxidant and growth regulator in plants [18]. It is a dominant cofactor for carbon transfer functions that are involved in various cellular reactions such as the synthesis of purines, photorespiration cycle, chlorophyll, and choline [19]. FA plays a crucial biochemical role in nucleic acid synthesis and amino acid metabolism and has become a common source of B complex vitamins [20]. Accordingly, it can be used as a novel, efficient organic fertilizer to boost the physiological processes of plants and conserve nutrients. It enhances solubilizing micronutrients, cell elongation, water retention, and cation exchange capacity [21]. Furthermore, it supports the transfer of amino acids to the proper location in protein chain creation and is also involved in the methylation of amino acids [22]. It has valuable impacts on catching the active oxygen that is generated during respiration and photosynthesis processes [22].

Ascorbic acid (AA) is involved in numerous processes such as cell division, cell expansion, cell enlargement, photoprotection, photosynthesis, flowering, final yield development, and carbohydrate synthesis [23]. It contributes to various metabolic and cell signaling activities. AA regulates numerous physiological functions controlling the development, growth, and tolerance of diverse environmental stresses [24]. It plays a decisive role in the photosynthesis process and adjusts the redox state of photosynthetic electron carriers [25]. AA is a main component in the ascorbate–glutathione cycle and reduces oxidative stress by regulating ROS detoxification [26].

Salicylic acid (SA) is a plant hormone produced naturally and belongs to the group of phenolic acids [27]. SA is considered an endogenous regulator in plants and plays a crucial role in several physiological and biological processes [28,29]. SA regulates the synthesis and signaling of other hormones such as ethylene, jasmonic acid, and auxin [30,31]. Moreover, it adjusts membrane permeability, stomatal conductivity, ion uptake, transport, and growth development [32]. Accordingly, it plays an integral role in improving plant tolerance to biotic and abiotic stresses [33].

Folic, ascorbic, and salicylic acids are affordable and convenient growth regulators to enhance plant efficiency, in particular under environmental stresses [34,35,36]. Accordingly, there is progress in assessing the roles of these acids on plants, but several aspects of physiological responses necessitate further research. Accordingly, in the present study, we aimed at evaluating the impact of foliar spray using three antioxidant substances, folic acid, ascorbic acid, and salicylic acid, on morphological, physiological, biochemical, agronomic, and quality traits in potato in salt-affected soil. Furthermore, we studied the association among the evaluated traits under salinity conditions.

## 2. Results

### 2.1. Vegetative Growth Attributes

Exogenous application using three safe–efficient substances, folic acid (FA), ascorbic acid (AA), and salicylic acid (SA), at different concentrations exhibited positive significant effects (*p* ≤ 0.05) on potato growth attributes such as plant height, shoot fresh and dry weight, and plant leaves area compared with untreated plants in salt-stressed soil (Table 1). The foliar-supplied AA at a rate of 200 mg/L displayed the best performance of growth attributes. It increased plant height by 56.2%, shoot fresh weight by 147.3%, shoot dry weight by 153.8%, and plant leaves area by 149.2% compared to untreated plants under salinity stress (averaged over two seasons).

### 2.2. Photosynthetic Pigments and Gas Exchange

The foliar treatments, i.e., folic acid (FA), ascorbic acid (AA), and salicylic acid (SA), had significant positive impacts (*p* ≤ 0.05) on photosynthetic pigments (i.e., chlorophyll a and b and carotenoids) and gas exchange (*Pn, Tr,* and *gs*) of potato plants under salt-affected soil compared to untreated control (Table 2). The best treatment was AA at the rate of 200 mg/L, which improved chlorophyll a by 52.9%, chlorophyll b by 48.6%, carotenoids by 13.4%, *Pn* by 79.1%, *Tr* by 80.9%, and *gs* by 86.8% compared to untreated control over the two seasons.

### 2.3. Nutrient Content and K^+^/Na^+^ Ratio

All foliar treatments displayed a significant effect (*p* ≤ 0.05) on the nutrient content of N, P, and K^+^ and the ratio of K^+^/Na^+^ while substantially decreasing Na^+^ and Cl^−^ content (Table 3). The application of AA at a rate of 200 mg/L exhibited the highest increase in the content of N, P, K and the ratio of K^+^/Na^+^ by 75.2, 13.6, 81.6, and 159.2% while decreasing Na^+^ and Cl^−^ by 62.3 and 67.7%, respectively, compared to untreated plants under salt-stressed soil over the two seasons.

### 2.4. Osmoprotectant and Antioxidant Enzymes

All foliar treatments (i.e., FA, SA, and AA at different rates) exhibited a significant impact (*p* ≤ 0.05) on the activity of osmoprotectants (i.e., proline and soluble sugars) and all enzymatic antioxidants (i.e., CAT, POD, APX, and SOD) compared to untreated plants (Table 4). The highest osmoprotectant and enzymatic activities were obtained using the application of AA at a rate of 200 mg/L in both seasons. It enhanced proline, soluble sugars, CAT, POD, APX, and SOD by 37.2, 50.6, 15.7, 55.7, 56.7, and 16.2%, respectively, compared to untreated control under salt-affected soil.

### 2.5. Yield and Its Components

Applied foliar application using FA, AA, and SA at different concentrations exhibited a significant increase (*p* ≤ 0.05) in yield grade 1, grade 2, grade 3, marketable yield, and total yield in both seasons (Table 5). Foliar treatment of AA at rates of 200 and 100 mg/L recorded the highest marketable and total yield followed by FA at rates of 150 and 100 mg/L.

### 2.6. Tuber Quality

Tuber dry matter, protein, starch percentage, total carbohydrates, and sugar percentage substantially increased as a result of using foliar applications of FA, AA, and SA compared with untreated plants under salinity stress (Table 6). The highest value of tuber quality was obtained by AA followed by FA and SA. The application of AA at rates 200 and 100 mg/L exhibited the uppermost values of tuber quality. It increased DM by 33.3%, protein by 156.6%, starch by 40.5%, total carbohydrates by 40.9%, total sugar by 58.5%, reducing sugar by 43.3%, and nonreducing sugar by 65.7% compared to untreated plants (averaged over two seasons).

## 3. Discussion

Soil salinity reduces water availability to plant roots; decreases metabolic functions, photosynthetic capacity, and cell elongation; and causes nutrient imbalance, ion toxicity, and overproduction of ROS [37]. These devastating impacts are reflected in considerable reductions in potato growth and productivity. Accordingly, it is valuable to assess the impact of exogenously applied antioxidant materials to attenuate the detrimental impacts on potato tuber yield and quality under salinity stress conditions. In the current study, three safe–efficient substances, folic acid, ascorbic acid, and salicylic acid, were applied in different concentrations to investigate their influence on growth, physio-biochemical, yield, and quality traits under salt-stressed soil.

The obtained results revealed that the exogenous application of ascorbic acid (AA) at rates of 200 and 100 mg/L exhibited the uppermost enhancement of all evaluated parameters under salinity stress. The application of AA significantly improved photosynthetic pigments, gas exchange, and nutrient content. This enhancement could be owing to the functions of AA as an essential cofactor of various enzymes or protein complexes [23,38]. This is in harmony with El-Hifny and El-Sayed [39], who manifested that AA treatment stimulated photosynthetic pigments and soluble solid substances in plant leaves compared with untreated plants under saline conditions. Moreover, Midan and Sorial [40] demonstrated superior impacts of AA foliar application on total chlorophyll, total phenols, total carbohydrates, and percentage of N, P, and K in lettuce. Shalaby and El-Ramady [41] disclosed that AA application improved the accumulation of N, P, and K and referred to its positive impact on nutrient uptake. Additionally, AA enhanced the osmoprotectants proline and soluble sugars and the antioxidant enzymes CAT, POD, APX, and SOD (Table 4). This impact could be owing to the effect of AA, which is a water-soluble antioxidant that regulates several physiological processes controlling plant development [24,42]. Furthermore, it participates in different metabolic processes and has a dynamic relationship with reactive oxygen species (ROS) [43,44,45]. In this context, Sajid and Aftab [46] elucidated that the activity of antioxidant enzymes SOD, POD, CAT, and APX increased in potato substantially under salinity stress conditions by applied foliar ascorbic acid. All aforementioned improvements are reflected in boosting plant growth, tuber yield, and tuber quality compared with untreated plants under salinity stress.

Folic acid (FA, vitamin B9) followed AA in its positive impacts on potato growth, tuber yield, and quality. The obtained results elucidated that FA foliar spray displayed a positive effect and significant increase in all vegetative growth parameters compared with untreated plants under salinity stress. FA exogenous application increased growth parameters by enhancing photosynthetic pigments, the metabolism of amino acids, methionine synthesis, and the photorespiration cycle, which led to improving potato growth [47]. Furthermore, FA has an important impact on plant growth auxin activity that contributes to improving dry matter [48]. Similarly, Ibrahim et al. [49] disclosed that FA application significantly increased growth parameters in potato, including plant length, leaf area, dry weight, and chlorophyll content compared with untreated potato plants. Additionally, FA application boosted photosynthetic pigments and nutrient content (N, P, and K) compared with untreated plants. Tuber yield and quality were significantly increased in the two growing seasons. These results are in harmony with those reported by Babarabie et al. [50], who elucidated that the applied foliar FA on improved chlorophyll content, photosynthetic rate, and plant productivity. Moreover, Cordeiro et al. [51] elucidated that FA enhanced the activities of enzymatic antioxidants compared to untreated controls. Likewise, Wang et al. [52] and Li et al. [53] disclosed that FA ameliorated POD, SOD, CAT, and APX activities under abiotic stresses.

Salicylic acid (SA) ranked third in terms of positive impact on potato growth, yield, and quality. Its exogenous application considerably enhanced photosynthetic pigments, gas exchange, nutrient content, enzymatic and nonenzymatic antioxidants. This enhancement could be due to the effect of SA as a growth regulator cofactor that boosts the photosynthesis process and synthesis of several growth-stimulating hormones such as cytokinin and auxin, which enhance cell elongation and division [54,55]. Therefore, its exogenous application efficiently enhanced vegetative growth, photosynthetic ability, fruit development, and, finally, boosted plant productivity [56,57]. In this context, Fariduddin et al. [58] proved that SA enhanced net photosynthetic rate, nitrogen metabolism, the activity of nitrate reductase, carboxylation efficiency, and seed yield. Moreover, Yıldırım and Dursun [59] demonstrated that SA foliar application improved photosynthetic pigments, mineral content, plant height, shoot and root fresh weight, shoot and root dry weight, shoot diameter, and leaf number/plant in tomato. Sahu and Sabat [60] deduced that exogenous application of SA organized the activities of intracellular antioxidant enzymes and accelerated plant tolerance to salinity stress. Likewise, Liu et al. [61] disclosed that the activities of CAT, POX, and SOD were enhanced using the exogenous application of SA under salinity stress.

## 4. Materials and Methods

### 4.1. Experimental Site

A two-year field experiment was carried out during the 2020 and 2021 growing seasons at the area designated for potato production at El-Noubaria, Egypt (30°43′54″ N; 30°33′01″ E). The soil samples were collected from the study site before planting in the two seasons and analyzed according to Black [62] and Jackson [63], as presented in Table 7. The soil analysis indicates saline soil, since the soil EC was 7.11 and 7.17 dS/m in the two seasons, respectively [64].

### 4.2. Agricultural Practices

A drip irrigation system was installed for the experiment with a distance between laterals and emitters of 0.80 m and 0.30 m, respectively. The plot comprised four rows with a length of 4 m long, and the plots were split by two empty rows. The potato seeds were planted at a distance of 0.30 m. The emitters’ flow rate was 4 L per hour, and the total applied water was 7400 m^3^/ha for each season. Nitrogen, phosphorus, and potassium fertilizers were used such as ammonium sulfate (20.5% N) at 285 kg/ha, calcium superphosphate (15.5% P_2_O_5_) at 180 kg/ha, and potassium sulfate (48% K_2_O) at 230 kg/ha, respectively. Phosphorus fertilizer was applied before planting with one-third of the amounts of N and K in the middle of the ridge then covered by sand. In addition, FYM was added before planting at a rate of 70 m^3^/ha. The remaining amounts of N and K (two-thirds) were divided into equal portions and added as soil application at 15-day intervals following complete emergence. Standard agriculture practices were performed as recommended for the commercial production of potato. According to the optimal period for growing potato in Egypt, planting was conducted in the first week of January for both seasons. The used potato seed tubers were *Solanium tuberosum* L. cv. Spunta, which is an important exported cultivar to European markets. The tubers were harvested by hand at the end of April.

### 4.3. Foliar Application

Folic acid (FA), ascorbic acid (AA), and salicylic acid were obtained from Sigma Aldrich Company. Foliar spray of FA (C_19_H_10_N_7_O_6_, MW: 441.4) was performed at rates of 100 and 150 mg/L. Foliar spray of AA (C_6_H_8_O_6_ MW: 176.12) was conducted at rates of 100 and 200 mg/L. Foliar spray of SA (2-hydroxybenzene-carboxylic, C_6_H_4_(OH)COOH, MW: 138.1) was implemented at rates of 100 and 200 mg/L. The used substances were sprayed using external spray over the plants’ leaves with a pressurized spray bottle with 0.1% Tween 20 as the surface spreader. Each plot received 2 L of aqueous solutions of safety substances 45, 60, and 75 days after planting. The control plants (nontreated) were sprayed with water and a spreading agent only.

### 4.4. Vegetative Growth Attributes

Five plants were collected randomly from the two external rows of each experimental plot at 80 days after planting to estimate shoot fresh and dry weight (g), leaf area/plant (cm^2^), and plant height (cm). Leaf area/plant (cm^2^) = (leaves dry weight (gm) × disk area)/(disk dry weight (gm)). Leaves and shoots were oven-dried at 70 °C till constant weight, then their dry weight was obtained per plant.

### 4.5. Photosynthetic Pigments and Gas Exchange

The photosynthetic pigments, i.e., chlorophyll *a* and *b* and carotenoids, were extracted from fresh plant leaf samples utilizing pure acetone, as outlined by Fadeel [65]. The extracts were purified, and then the optical density of the filtrate was identified calorimetrically by applying the wavelengths of 662, 644, and 440.5 nm for chlorophyll *a* and *b* and carotenoids, respectively. The pigments as mg/g fresh weight were measured according to the formula adopted by von Wettstein [66]. Leaf net photosynthetic rate (*Pn*), rate of transpiration (*Tr*), and stomatal conductance (*gs*) were measured for photosynthetic parameters by employing a portable photosynthesis system (LF6400XTR, LI-COR, Lincoln, NE, USA). The measurements were performed at 09:00–11:00 a.m. on the fully expanded leaf.

### 4.6. Determination of Nutrient Content

For nutrient determination, a weight of 0.2 g of dried leaves was digested with 96% H_2_SO_4_ in the presence of H_2_O_2_ [67] and diluted with distilled water. Total nitrogen and total potassium were measured in the shoot utilizing the method of micro Kjeldahl and a flame photometer device in the same order following Chapman and Pratt [68]. The total phosphorus in the shoot was measured by the colorimetric method using the ascorbic acid method [69]. Na^+^ and K^+^ content was measured using a flame photometer [70], and the content of leaf Cl^−^ was assessed [71].

### 4.7. Determination of Osmoprotectant Components

Proline accumulation in potato leaves was determined following the method of Bates et al. [72]. A 0.1 g amount of fresh leaf material was ground with 10 mL of 3% (*w*/*v*) aqueous sulfosalicylic acid. The homogenate was filtered using Whatman 2 filter paper. One milliliter of the filtrate was reacted with 1 mL of acid ninhydrin reagent and 1 mL of glacial acetic acid in a test tube for 1 h at 100 °C, and the reaction terminated in an ice bath. Two milliliters of toluene was added to the mixture, and the absorbance of the upper toluene layer was recorded at 520 nm using a UV spectrophotometer.

Total soluble sugar content was assessed as follows: 0.2 g of leaves was rinsed with 5 mL of 70% ethanol and homogenized with 5 mL of 96% ethanol. The extract was centrifuged at 3500 × g for 10 min. The supernatant was collected and stored at 4 °C [73]. Freshly prepared enthrone (3 mL) was added to 0.1 mL of supernatant. This mixture was incubated in a hot water bath for 10 min. The absorbance was recorded at 625 nm with a Bausch and Lomb-2000 Spectronic Spectrophotometer.

### 4.8. Determination of Antioxidants Activities

The enzymes were extracted according to Vitória et al. [74]. The concentration of catalase (CAT) enzyme was measured spectro-photochemically according to Chance and Maehly [75]. Peroxidase (POD) activity was estimated according to Thomas et al. [76]. Ascorbate peroxidase (APX) was measured spectro-photochemically according to Fielding and Hall (1978). The activity of superoxide dismutase (SOD) was measured by recording the drop in absorbance of the superoxide-nitro blue tetrazolium complex by the enzyme [77].

### 4.9. Yield and Its Components

One hundred and ten days from planting, tubers from each experimental unit were harvested, counted, weighed, and graded into three sizes according to their diameters (above 6.5 cm, 3.5–6.4 cm, and less than 3.5 cm). Then, each grade was weighed separately, and average tuber weight and tuber yield per ha were calculated.

### 4.10. Tuber Quality

One hundred grams of the grated mixture was dried at 105 °C until constant weight, and then dry matter percentage (DM%) was measured. The percentage of total protein was calculated by multiplying total N by 6.25. Starch percentage was computed following the equation of Burton [78]: starch (%) = 17.546 + 0.891 (tuber DM%—24.18). Carbohydrate percentage was measured in the dried samples of tubers for all treatments photometrically according to Bernfeld [79] and Miller [80] methods. Total, reducing, and nonreducing sugar percentages were determined in the dried samples of tubers of all treatments photometrically according to the methods of Bernfeld [79] and Miller [80].

### 4.11. Statistical Analysis

R software was used to statistically analyze all data of this study. Analysis of variance (ANOVA) was performed, and Shapiro–Wilk and Bartlett’s tests were applied to check the normality distribution of the residuals and homogeneity of variances, respectively. Differences among all treatments were separated by the least significant difference (LSD) at *p* ≤ 0.05.

## 5. Conclusions

Exogenously sprayed ascorbic acid (AA) at rates of 200 and 100 mg/L substantially mitigated the deleterious effects of salinity stress and stimulated plant growth by ameliorating all of the assessed physio-biochemical parameters. Furthermore, folic acid (FA) at rates of 150 and 100 mg/L, as well as salicylic acid (SA) at rates of 200 and 100 mg/L, considerably alleviated the detrimental impacts of salinity stress. The application of antioxidant materials (FA, AA, and SA) separately enhanced the activity of osmoprotectants and the enzymatic defense system components under salinity stress. Therefore, it could be concluded that these materials had a positive impact on the physiological, biochemical, tuber yield, and quality traits of salt-stressed potato plants.

## Figures and Tables

**Table 1 plants-11-00210-t001:** Impact of exogenously sprayed folic acid in two rates of 100 and 150 mg/L, ascorbic acid in two rates of 100 and 200 mg/L, and salicylic acid in two rates of 100 and 200 mg/L compared with untreated control on plant height, fresh weight of shoots, dry weight of shoots, and leaves area per plant potato in salt-affected soil over two growing seasons.

Treatment	Plant Height (cm)	Fresh Weight of Shoots Plant (g)	Dry Weight of Shoots Plant (g)	Leaves Area/Plant (cm^2^)
Untreated control	51.1 ± 2.2 ^f^	131.6 ± 3.8 ^f^	16.6 ± 1.0 ^g^	1356 ± 5.9 ^f^
Folic acid 100 mg/L	64.3 ± 2.9 ^d^	349.3 ± 5.2 ^c^	43.5 ± 1.6 ^d^	3337 ± 6.1 ^c^
Folic acid 150 mg/L	67.3 ± 2.8 ^c^	364.8 ± 4.7 ^bc^	45.1 ± 1.4 ^c^	3897 ± 7.6 ^b^
Ascorbic acid 100 mg/L	72.1 ± 3.2 ^b^	386.3 ± 4.6 ^b^	47.8 ± 1.7 ^b^	4500 ± 8.4 ^a^
Ascorbic acid 200 mg/L	79.6 ± 3.0 ^a^	455.3 ± 5.3 ^a^	58.5 ± 2.2 ^a^	4734 ± 8.2 ^a^
Salicylic acid 100 mg/L	61.0 ± 2.9 ^e^	247.5 ± 3.8 ^e^	30.8 ± 1.0 ^f^	2632 ± 5.8 ^e^
Salicylic acid 200 mg/L	63.6 ± 2.6 ^d^	311.8 ± 3.4 ^d^	38.8 ± 1.4 ^e^	3043 ± 7.8 ^d^

Means followed by different letters differ significantly by LSD (*p* < 0.05).

**Table 2 plants-11-00210-t002:** Impact of exogenously sprayed folic acid in two rates of 100 and 150 mg/L, ascorbic acid in two rates of 100 and 200 mg/L, and salicylic acid in two rates of 100 and 200 mg/L compared with untreated control on chlorophyll *a* (Chl_a_), chlorophyll *b* (Chl_b_), carotenoids, net photosynthetic rate (*Pn*), transpiration rate (*Tr*), and stomatal conductance (*Gs*) of potato in salt-affected soil over two growing seasons.

Treatment	Chl_a_ (mg g^−1^ *F*W)	Chl_b_ (mg g^−1^ *F*W)	Carotenoids (mg g^−1^ *F*W)	*Pn* (µmol CO_2_ m^−2^ s^−1^)	*Tr* (mmol H_2_O m^−2^ s^−1^)	*gs* (mmol H_2_O m^−2^ s^−1^)
Untreated control	0.97 ± 0.03 ^e^	0.70 ± 0.03 ^d^	0.90 ± 0.03 ^d^	5.46 ± 0.13 ^g^	3.75 ± 0.11 ^d^	0.305 ± 0.02 ^d^
Folic acid 100 mg/L	1.27 ± 0.05 ^c^	0.84 ± 0.04 ^bc^	0.97 ± 0.04 ^abc^	9.10 ± 0.18 ^d^	5.13 ± 0.13 ^b^	0.423 ± 0.03 ^b^
Folic acid 150 mg/L	1.31 ± 0.07 ^c^	0.88 ± 0.03 ^b^	0.98 ± 0.03 ^abc^	9.51 ± 0.26 ^c^	5.25 ± 0.12 ^b^	0.513 ± 0.02 ^a^
Ascorbic acid 100 mg/L	1.38 ± 0.09 ^b^	1.03 ± 0.05 ^a^	1.00 ± 0.06 ^ab^	10.05 ± 0.24 ^b^	6.33 ± 0.15 ^a^	0.536 ± 0.04 ^a^
Ascorbic acid 200 mg/L	1.48 ± 0.07 ^a^	1.04 ± 0.04 ^a^	1.02 ± 0.05 ^a^	10.70 ± 0.30 ^a^	6.78 ± 0.14 ^a^	0.563 ± 0.03 ^a^
Salicylic acid 100 mg/L	0.98 ± 0.04 ^e^	0.75 ± 0.02 ^cd^	0.93 ± 0.07 ^cd^	7.19 ± 0.18 ^f^	4.35 ± 0.12 ^c^	0.385 ± 0.01 ^c^
Salicylic acid 200 mg/L	1.15 ± 0.08 ^d^	0.80 ± 0.03 ^c^	0.95 ± 0.04 ^bcd^	7.64 ± 0.17 ^e^	5.04 ± 0.13 ^b^	0.413 ± 0.02 ^b^

Means followed by different letters differ significantly by LSD (*p* < 0.05).

**Table 3 plants-11-00210-t003:** Impact of exogenously sprayed folic acid in two rates of 100 and 150 mg/L, ascorbic acid in two rates of 100 and 200 mg/L, and salicylic acid in two rates of 100 and 200 mg/L compared with untreated control on the concentration of nutrients and the ratio of K^+^/Na^+^ of potato in salt-affected soil over two growing seasons.

Treatment	Concentration of Nutrients (%)					K^+^/Na^+^ Ratio
	N	P	K	Na	Cl	
Untreated control	2.28 ± 0.09 ^f^	0.370 ± 0.02 ^e^	2.09 ± 0.11 ^e^	0.820 ± 0.03 ^a^	0.745 ± 0.05 ^a^	2.64 ± 0.10 ^f^
Folic acid 100 mg/L	2.98 ± 0.10 ^d^	0.405 ± 0.03 ^b^	3.21 ± 0.12 ^c^	0.516 ± 0.01 ^c^	0.446 ± 0.02 ^c^	6.23 ± 0.21 ^d^
Folic acid 150 mg/L	3.19 ± 0.07 ^c^	0.411 ± 0.02 ^b^	3.28 ± 0.13 ^c^	0.420 ± 0.02 ^d^	0.353 ± 0.02 ^d^	7.86 ± 0.32 ^c^
Ascorbic acid 100 mg/L	3.50 ± 0.09 ^b^	0.421 ± 0.03 ^a^	3.52 ± 0.15 ^b^	0.363 ± 0.02 ^de^	0.293 ± 0.02 ^de^	9.75 ± 0.33 ^b^
Ascorbic acid 200 mg/L	3.99 ± 0.15 ^a^	0.427 ± 0.03 ^a^	3.79 ± 0.11 ^a^	0.313 ± 0.01 ^e^	0.243 ± 0.01 ^e^	12.15 ± 0.41 ^a^
Salicylic acid 100 mg/L	2.38 ± 0.06 ^f^	0.380 ± 0.01 ^d^	2.80 ± 0.12 ^d^	0.668 ± 0.04 ^b^	0.608 ± 0.04 ^b^	4.20 ± 0.11 ^e^
Salicylic acid 200 mg/L	2.79 ± 0.07 ^e^	0.392 ± 0.03 ^c^	2.88 ± 0.11 ^d^	0.605 ± 0.04 ^c^	0.527 ± 0.03 ^bc^	4.77 ± 0.17 ^e^

Means followed by different letters differ significantly by LSD (*p* < 0.05).

**Table 4 plants-11-00210-t004:** Impact of exogenously sprayed folic acid in two rates of 100 and 150 mg/L, ascorbic acid in two rates of 100 and 200 mg/L, and salicylic acid in two rates of 100 and 200 mg/L compared with untreated control on free proline, soluble sugars, catalase (CAT), peroxidase (POX), superoxide dismutase (SOD), and ascorbate peroxidase (APX) of potato in salt-affected soil over two growing seasons.

Treatment	Free Proline (µmol/g DW)	Soluble Sugars (mg/g DW)	CAT	POX	SOD	APX
			A564 min^−1^ mg^−1^ Protein			
Untreated control	29.1 ± 1.5 ^f^	21.1 ± 1.4 ^g^	65.8 ± 3.1 ^f^	1.06 ± 0.10 ^e^	5.16 ± 0.16 ^e^	56.0 ± 2.7 ^f^
Folic acid 100 mg/L	34.0 ± 2.6 ^cd^	26.1 ± 1.6 ^d^	70.4 ± 3.5 ^d^	1.35 ± 0.12 ^c^	6.90 ± 0.19 ^cd^	61.9 ± 2.6 ^d^
Folic acid 150 mg/L	35.6 ± 2.8 ^c^	27.5 ± 1.7 ^c^	72.1 ± 3.1 ^c^	1.46 ± 0.13 ^b^	7.36 ± 0.23 ^bc^	63.3 ± 3.0 ^c^
Ascorbic acid 100 mg/L	37.4 ± 2.3 ^b^	29.1 ± 1.6 ^b^	73.6 ± 3.6 ^b^	1.55 ± 0.14 ^b^	7.68 ± 0.28 ^ab^	64.5 ± 3.5 ^b^
Ascorbic acid 200 mg/L	39.9 ± 2.7 ^a^	31.7 ± 1.5 ^a^	76.1 ± 3.7 ^a^	1.65 ± 0.14 ^a^	8.08 ± 0.29 ^a^	65.1 ± 3.4 ^a^
Salicylic acid 100 mg/L	31.5 ± 2.6 ^e^	23.1 ± 1.6 ^f^	67.5 ± 3.6 ^e^	1.19 ± 0.15 ^d^	6.40 ± 0.21 ^d^	60.1 ± 3.3 ^e^
Salicylic acid 200 mg/L	33.3 ± 2.5 ^d^	24.7 ± 1.5 ^e^	69.4 ± 3.7 ^d^	1.29 ± 0.12 ^c^	6.63 ± 0.20 ^d^	60.7 ± 3.8 ^e^

Means followed by different letters differ significantly by LSD (*p* < 0.05).

**Table 5 plants-11-00210-t005:** Impact of exogenously sprayed folic acid in two rates of 100 and 150 mg/L, ascorbic acid in two rates of 100 and 200 mg/L, and salicylic acid in two rates of 100 and 200 mg/L compared with untreated control on tuber yield traits of potato in salt-affected soil over two growing seasons.

Treatment	Yield of Grade 1 (ton/ha)	Yield of Grade 2 (ton/ha)	Yield of Grade 3 (ton/ha)	Marketable Yield (ton/ha)	Total Yield (ton/ha)
Untreated control	25.2 ± 1.5 ^e^	2.62 ± 0.07 ^f^	0.33 ± 0.02 ^e^	27.8 ± 1.1 ^f^	28.1 ± 1.4 ^g^
Folic acid 100 mg/L	38.6 ± 2.2 ^c^	10.16 ± 0.04 ^cd^	1.25 ± 0.04 ^c^	48.7 ± 1.7 ^d^	50.0 ± 2.2 ^d^
Folic acid 150 mg/L	43.7 ± 2.7 ^b^	10.70 ± 0.20 ^c^	1.24 ± 0.05 ^c^	54.4 ± 2.3 ^c^	55.7 ± 2.5 ^c^
Ascorbic acid 100 mg/L	43.5 ± 2.9 ^b^	11.60 ± 0.19 ^b^	1.19 ± 0.05 ^b^	55.3 ± 3.1 ^b^	56.5 ± 3.1 ^b^
Ascorbic acid 200 mg/L	49.8 ± 2.7 ^a^	14.10 ± 0.26 ^a^	1.53 ± 0.07 ^a^	62.9 ± 2.5 ^a^	64.5 ± 3.2 ^a^
Salicylic acid 100 mg/L	33.5 ± 1.2 ^d^	7.50 ± 0.15 ^e^	0.76 ± 0.03 ^d^	41.0 ± 2.1 ^e^	41.8 ± 2.0 ^f^
Salicylic acid 200 mg/L	38.7 ± 1.5 ^c^	9.42 ± 0.18 ^d^	1.21 ± 0.05 ^c^	48.1 ± 2.6 ^d^	49.3 ± 2.5 ^e^

Means followed by different letters differ significantly by LSD (*p* < 0.05).

**Table 6 plants-11-00210-t006:** Impact of exogenously sprayed folic acid in two rates of 100 and 150 mg L^−1^, ascorbic acid in two rates of 100 and 200 mg L^−1^, and salicylic acid in two rates of 100 and 200 mg/L compared with untreated control on tuber quality of potato tubers in salt-affected soil over two growing seasons.

Treatment	Dry Matter (%)	Protein (%)	Starch (%)	Total Carbohydrates (%)	Sugars (%)		
					Total	Reducing	Non- Reducing
Untreated control	19.1 ± 1.3 ^d^	6.21 ± 0.3 ^e^	13.4 ± 0.6 ^d^	58.0 ± 1.6 ^f^	5.36 ± 0.3 ^f^	1.71 ± 0.05 ^f^	3.65 ± 0.1 ^e^
Folic acid 100 mg/L	21.9 ± 1.4 ^c^	10.4 ± 0.5 ^c^	15.9 ± 0.8 ^c^	75.6 ± 2.7 ^cd^	6.75 ± 0.4 ^c^	1.99 ± 0.04 ^c^	4.74 ± 0.2 ^b^
Folic acid 150 mg/L	23.4 ± 1.2 ^b^	11.1 ± 0.6 ^bc^	17.4 ± 0.7 ^b^	76.6 ± 2.8 ^c^	6.84 ± 0.3 ^bc^	2.03 ± 0.06 ^bc^	4.82 ± 0.2 ^b^
Ascorbic acid 100 mg/L	25.0 ± 1.5 ^a^	12.3 ± 0.7 ^b^	18.6 ± 0.8 ^a^	79.5 ± 3.3 ^b^	6.93 ± 0.6 ^b^	2.07 ± 0.03 ^b^	4.86 ± 0.3 ^b^
Ascorbic acid 200 mg/L	25.4 ± 1.6 ^a^	15.9 ± 0.9 ^a^	18.8 ± 0.7 ^a^	81.7 ± 3.1 ^a^	8.49 ± 0.5 ^a^	2.45 ± 0.07 ^a^	6.04 ± 0.4 ^a^
Salicylic acid 100 mg/L	19.8 ± 1.2 ^d^	7.32 ± 0.4 ^d^	13.5 ± 0.6 ^d^	65.5 ± 2.0 ^e^	6.18 ± 0.2 ^e^	1.86 ± 0.04 ^e^	4.31 ± 0.2 ^d^
Salicylic acid 200 mg/L	20.9 ± 1.1 ^c^	7.88 ± 0.4 ^d^	15.6 ± 0.5 ^c^	74.8 ± 2.8 ^d^	6.45 ± 0.3 ^d^	1.95 ± 0.05 ^d^	4.50 ± 0.3 ^c^

Means followed by different letters differ significantly by LSD (*p* < 0.05).

**Table 7 plants-11-00210-t007:** Physical and chemical properties of the soil at the experimental site.

Soil Characteristic	1st Season	2nd Season
Soil particles distribution		
Sand (%)	45.66	45.43
Silt (%)	29.89	30.02
Clay (%)	24.45	24.55
Textural class	Loam	Loam
Field capacity, %	16.7	16.7
Calcium carbonate (CaCO_3_, g/kg)	62.0	60.4
Organic matter (g/kg)	7.98	7.65
pH	7.71	7.62
Electrical conductivity (EC, dS/m)	7.17	7.11
Soluble cations and anions (mmolc/L)		
Calcium (Ca^2+^)	16.7	16.6
Magnesium (Mg^2+^)	19.7	19.0
Sodium (Na^+^)	18.0	18.1
Potassium (K^+^)	6.47	6.69
Carbonate (CO_3_^2−^)	-	-
Bicarbonate (HCO_3_^−^)	20.7	20.9
Chloride (Cl^–^)	32.7	31.7
Sulfate (SO_4_^2−^)	8.4	8.19
Available nutrient (mg/kg soil)		
Nitrogen (N)	58.4	58.0
Phosphorus (P)	8.90	8.80
Potassium (K)	99.0	97.7

## Data Availability

The data presented in this study are available upon request from the corresponding author.

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
