# Peer review of "Impact of Exogenously Sprayed Antioxidants on Physio-Biochemical, Agronomic, and Quality Parameters of Potato in Salt-Affected Soil"

_plants, 2022, doi:10.3390/plants11020210_

Round 1

Reviewer 1 Report

In this manuscript the authors report the results of an investigation related to the positive impacts of foliar spray using; folic, ascorbic, and salicylic acids at different concentrations on potato growth in salt-affected soil. The study has been carefully carried out comparing the results obtained in each case with untreated control plants. These results are encouraging but the following items should be taken in consideration:

  1. The cost of this exogenous treatments using these acids has not been mentioned anywhere in the article and it should be since these acids are not easily affordable.
  2. In the conclusion, it’s a little bit confusing because it seems like the plants were submitted to a consecutive treatment with the three acids one after the other whereas in the rest of the article, the results seem to reflect the effect of every acid alone.
  3. Have you considered a foliar spray with a mixture of the three acids?
  4. Another control experiment should be pointed out in which a non-salt-affected soil is used in this experiment to check if the improvement is independent on the soil nature.

Additionally, the manuscript is written in an acceptable English which can be improved for the last version, see some of the mistakes found matched in red in the PDF enclosed. As a conclusion, I recommend publication subject to the minor amendments and clarifications mentioned above. 

Author Response

Responses to Reviewers Comments

Reviewer 1:

In this manuscript the authors report the results of an investigation related to the positive impacts of foliar spray using; folic, ascorbic, and salicylic acids at different concentrations on potato growth in salt-affected soil. The study has been carefully carried out comparing the results obtained in each case with untreated control plants. These results are encouraging but the following items should be taken into consideration:

Re: We would like to thank the Reviewer for his time dedicated to our manuscript and his/her positive assessment of our work.

The cost of these exogenous treatments using these acids has not been mentioned anywhere in the article and it should be since these acids are not easily affordable.

Re: The used acids are easily affordable in Egypt as well as in other countries as reported in published studies as Farjam et al. 2014, J. Appl. Bot. Food Qual., 87:80-86; Utami et al. 2018, Front. Chem. 6: 251; Khan et al. 2021, Int. J. Phytoremediation, 1-14;  more information has been added in lines 82-83.

In the conclusion, it’s a little bit confusing because it seems like the plants were submitted to a consecutive treatment with the three acids one after the other whereas in the rest of the article, the results seem to reflect the effect of every acid alone.

Re: The conclusion has been rewritten (lines 385-368)

Have you considered a foliar spray with a mixture of the three acids?

Re: No, we assessed the impacts of tested acids separately

Another control experiment should be pointed out in which a non-salt-affected soil is used in this experiment to check if the improvement is independent on the soil nature.

Re: The experimental site is located in a salt-affected region and there is no close non-salt-affected soil. Moreover, we mainly focused on the influence of the foliar application of studied acids at different concentrations on potato growth, physio-chemical ingredients, antioxidant defense system, and tuber yield and quality. Furthermore, we compared the evaluated treatments with untreated control which was sprayed with only water to check the enhancement due to used treatments under salinity conditions.

Additionally, the manuscript is written in acceptable English which can be improved for the last version, see some of the mistakes found matched in red in the PDF enclosed. As a conclusion, I recommend publication subject to the minor amendments and clarifications mentioned above.

Re: Thanks so much for your assistance, all comments in the PDF have been considered in the revised version (lines, 130, 221, 259, 266, 267, 288, 320, 325, and 343).

Reviewer 2 Report

This peer-reviewed scholarly publication presents interesting, complementary information on

positive effect of antioxidant application on physico-biochemical, agronomic and quality parameters of potato in saline soil.

The publication should be completed and corrected:

  1. The authors conducted a two-year experiment and the results should primarily be reported as averages over two years.
  2. The content from the introduction and discussion are repeated (characterization of the effect of tested antioxidants on physiological parameters of plants - their role in the plant) should be characterized in detail in the introduction. In the discussion, the focus should be on comparing the obtained results with other published results and determining the reason.
  3. chapter statistical analysis is insufficiently well described.

To be completed:

- what test was used to check normal distribution and homogeneity of variance

- what test was used to create homogeneous groups and evaluate the results.

  1. Supplement methods of investigating chemical properties of soils.
  2. Less significant comments are included in the text of the manuscript.

Author Response

Responses to Reviewers Comments

Reviewer: 2

This peer-reviewed scholarly publication presents interesting, complementary information on positive effect of antioxidant application on physico-biochemical, agronomic and quality parameters of potato in saline soil. The publication should be completed and corrected:

Re: We would like to thank the Reviewer for his time devoted to our manuscript and presenting positive aspects of our manuscript.

The authors conducted a two-year experiment and the results should primarily be reported as averages over two years.

Re: All presented results have been presented as averages over two years as suggested (Tables 1-6)

The content from the introduction and discussion are repeated (characterization of the effect of tested antioxidants on physiological parameters of plants - their role in the plant) should be characterized in detail in the introduction. In the discussion, the focus should be on comparing the obtained results with other published results and determining the reason.

Re: The introduction and discussion sections have been revised and the repeated parts have been removed. Following the Reviewer's suggestion, the discussion focused mainly on the obtained results and their harmony with published studies and determining the reason.

chapter statistical analysis is insufficiently well described. To be completed:

- what test was used to check normal distribution and homogeneity of variance

Re: Shapiro-Wilk and Bartlett’s tests were applied to check the normality distribution of the residuals and homogeneity of variances, respectively. More details have been added as suggested (lines 352-355).

Supplement methods of investigating chemical properties of soils.

Re: More details have been added, please see lines 265-261

Less significant comments are included in the text of the manuscript.

Re: The text has been revised and significant comments have been added, please see lines 100, 115, 128, 136, and 144.

Thanks so much for providing constructive suggestions to improve the quality of our manuscript.